# Supporting Occupational Physicians in the Implementation of Workers’ Health Surveillance: Development of an Intervention Using the Behavior Change Wheel Framework

**DOI:** 10.3390/ijerph18041939

**Published:** 2021-02-17

**Authors:** Felicia S. Los, Henk F. van der Molen, Carel T. J. Hulshof, Angela G. E. M. de Boer

**Affiliations:** Amsterdam Public Health Research Institute, Coronel Institute of Occupational Health, Department of Public and Occupational Health, University of Amsterdam, Amsterdam UMC, Meibergdreef 9, 1105 AZ Amsterdam, The Netherlands; h.f.vandermolen@amsterdamumc.nl (H.F.v.d.M.); c.t.hulshof@amsterdamumc.nl (C.T.J.H.); a.g.deboer@amsterdamumc.nl (A.G.E.M.d.B.)

**Keywords:** workers’ health surveillance, occupational physicians, behavior change, intervention development, training, education

## Abstract

Workers’ health surveillance (WHS) is an important preventive activity aimed at prevention of work-related diseases. However, WHS is not regularly implemented in some EU-countries. As occupational physicians (OPs) have to play an important role in implementation of WHS, this study aimed to develop an intervention to support OPs in implementation of WHS. The behavior change wheel framework (BCW) was used to develop the intervention. First, the problem was defined, and target behavior was selected by using results from a survey study among OPs. Subsequently, change objectives in target behavior were specified. Finally, appropriate intervention functions, behavior change techniques, and modes of delivery were identified to develop the intervention. Target behaviors were (1) OPs initiate WHS, and (2) OPs conduct preventive consultations with workers. OPs’ capabilities, and experienced opportunities were identified as change objectives. Intervention functions (education, training, enablement) and behavior change techniques (information about consequences, demonstration, instructions, behavioral practice, feedback on behavior, goal setting, action planning, reviewing goals) were selected to develop the intervention, delivered by face-to-face group training and e-learning. The proposed intervention consists of training and e-learning to support OPs in implementing WHS. Feasibility and effect of the intervention will be evaluated in future studies.

## 1. Introduction

To keep workers safe and healthy, preventive measures in the workplace aimed at the prevention of work-related diseases, as well as maintenance and promotion of workers’ health, are essential [1,2]. One important preventive activity is workers’ health surveillance (WHS) [3,4,5]. WHS consists of periodically monitoring workers to detect work-related risk factors for the deterioration of their health and work functioning. Furthermore, WHS aims to identify adverse health effects caused by work or working conditions (e.g., noise-induced hearing loss or work-related mental health disorders) and subsequently start appropriate preventive interventions if necessary [6]. Several studies have shown that the implementation of WHS, and subsequent preventive interventions can have a positive effect on workers’ health and work functioning [5,7,8] and can result in benefits for the employer (e.g., financial benefits) [9,10].

To ensure that workers receive WHS appropriate to the health and safety risks they are exposed to at their work, WHS is included in European regulation [4], while international technical and ethical guidelines have been developed for its implementation [3]. A review of the European regulation on WHS showed that, although the concept of health surveillance was not unequivocally defined and implemented in the countries of the EU, WHS is adopted by all EU countries [4]. Several studies showed that, in some countries, the activities of occupational physicians (OPs) are more focused on sickness absence and return-to-work activities than on executing preventive activities, such as WHS [11,12]. These findings indicate that WHS is not regularly implemented.

This is also the case in the Netherlands. Even though providing WHS to workers is a legal obligation for employers [13], the activities of OPs mainly focus on return to work [12]. WHS in the Netherlands comprises a voluntary medical examination of workers, discussion of the results with the worker, and advising and implementing interventions on the basis of the results. WHS can also lead to feedback and advice on preventive interventions on group level. To support OPs in their task to advise upon, set up, and carry out WHS at companies, a guidance document was developed and published by the Netherlands Society of Occupational Medicine (NVAB) [14]. According to this guidance document, WHS can have three core objectives: (1) prevention of onset or recurrence and/or monitoring of occupational and work-related diseases and injuries, (2) monitoring and promoting individuals’ health in relation to work, and (3) maintaining and improving the general health and sustainable employability of workers. For the employer to comply with the legal obligation, WHS should be aimed at the prevention of onset, recurrence and/or worsening of occupational and work-related diseases, at the least [13].

The implementation of preventive activities in the workplace may be difficult [15,16]. As multiple stakeholders are involved in the implementation of WHS, various different barriers for OPs need to be taken into consideration. Using the Behavior Change Wheel framework (BCW), it is possible to systematically identify barriers and develop an intervention for overcoming barriers to the implementation of WHS [17,18].

The BCW framework brings together several theory-based tools to understand and intervene in behavior change in target populations. According to this model, behavior is driven by three components: capability, opportunity, and motivation (COM-B components). Capability refers to a person’s psychological and physical capacity to engage in the target behavior. Opportunity refers to physical or psychological external factors that influence the potential success of the behavior. Motivation involves the psychological automatic and reflective processes that can trigger direct behavior. Analysis of the COM-B components can be used to determine what needs to change in the current behavior of the target population. After selecting the COM-B components that need to be changed in order to achieve the target behavior, the BCW framework provides intervention functions, policy categories, behavior change techniques, and modes of delivery to facilitate the development of the intervention.

The aim of this study is to develop an intervention for OPs aimed at overcoming the barriers to implementation of WHS using the eight-step BCW framework.

## 2. Materials and Methods

We followed the eight steps of the BCW framework to develop the intervention. The first part of the process was aimed at understanding the behavior. First the problem behavior was described in behavioral terms by listing all stakeholders and behaviors involved in the implementation of WHS. Second, the behavior to target in the intervention was identified from the list of behaviors. Third, the identified target behavior was specified, and, fourth, the COM-B components that needed change to achieve the desired behavior were selected by means of a nominal group discussion between the researchers. In the second part, the intervention content and implementation options were identified. Based on the designations of the BCW framework, literature and judgement of the researchers, the most appropriate intervention functions, policy categories, behavior change techniques (*BCTs*), and modes of delivery to target the COM-B components and achieve the desired behavior were identified. Finally, based on the selected intervention content and options, a proposal of an intervention was described.

### 2.1. Step 1: Define the Problem in Behavioral Terms

Defining the problem in behavioral terms means being specific about (1) the target group and (2) the behavior itself. A guidance document for OPs provided a ten-step description of the process of implementing WHS, including a description of OPs’ professional tasks and the role of other stakeholders involved in the process [14]. To define the problem in behavioral terms, stakeholders involved in the process of implementing WHS and the behaviors for the implementation of WHS, as described in the guidance document, were listed.

### 2.2. Step 2: Select the Target Behavior

For the selection of the target behaviors, the results from a previously conducted survey study on the implementation of WHS among OPs were used. The survey investigated OPs’ performance, capabilities, motivation, opportunities and needs in respect of the implementation of WHS (described in detail in Los et al., 2019 [19]). OPs’ desired behaviors for the implementation of WHS, as described in the guidance document, were compared with results about performance of WHS from the survey study among OPs to identify barriers in OPs’ behavior to the implementation of WHS. These behaviors were selected as behavior to target in the intervention.

### 2.3. Step 3: Specify the Target Behavior

The research team specified the target behavior selected in step 2 by defining who performs the behavior and what they need to do to achieve the desired behavior, where and when they need to do it, how often, and with whom.

### 2.4. Step 4: Identify What Needs to Change

In step 4, the research team identified what needs to change in the COM-B components of the target behavior to achieve the desired behavior. COM-B components for the desired behaviors were listed. The results from the survey study on capabilities, opportunities, and motivation from OPs for the implementation of WHS [19] were compared with the COM-B components for the desired behaviors to identify what needed to change.

To reach consensus about the change needed in the COM-B components, the nominal group technique was used [20,21] in a session with the four authors (FL, AB, CH, HM) involved in the development of the intervention. FL made a final overview of COM-B components and put forward reasons or arguments for the need to change them. This overview was presented to all authors three days before the session. The session was held face to face and consisted of five rounds (see Figure 1).

### 2.5. Step 5: Identify Intervention Functions

The BCW framework provides an overview of intervention functions, each linked to certain COM-B components [18]. First, a list of intervention functions linked to the COM-B components selected for change were listed. To identify the most appropriate intervention functions, each listed intervention function was assessed on the following criteria suggested by the BCW framework: (1) Affordability, (2) Practicability, (3) Effectiveness or cost-effectiveness, (4) Acceptability, (5) Side-effects or safety, and (5) Equity (APEASE) [18]. In a literature search in PubMed and the Cochrane database, limited to English-language articles published between January 2000 and April 2020, evidence was sought to support our judgement on the APEASE criteria. We searched for articles on training programs, educational programs, and intervention programs aiming to change the behavior of OPs and physicians in general, and examples of existing interventions for OPs in the Netherlands. When literature for the context of OPs or WHS to support a judgement on the APEASE criteria was lacking, the criteria were judged as unknown.

To reach consensus about the intervention functions to be included in the intervention, a group discussion was held within the research team. First, all intervention functions that met the criteria of affordability, practicability, acceptability, and effectiveness were included as they are prerequisites for implementing an intervention. Subsequently, through group discussion with all authors, consensus was reached about which intervention functions of which judgement on affordability, practicability, acceptability, or effectiveness was missing, would be most appropriate to change the selected COM-B components.

### 2.6. Step 6: Identify Policy Categories

The BCW framework suggests which policy categories are likely to be appropriate for supporting different intervention functions. For each intervention function, policy categories suggested in the guidance document were listed [18]. However, as changing policy was not the primary concern in this study, this step was not done in detail, other than listing the policy categories that may be relevant to the intervention.

### 2.7. Step 7: Identify Behavior Change Techniques

In step 7, the most appropriate BCTs for operationalizing the intervention functions selected in step 5 were identified. The BCTs most frequently used based on the links previously drawn between the BCW and the taxonomy of 93 BCTs were listed [18]. The most appropriate BCTs in terms of the APEASE criteria were selected. Again, a literature search in Pubmed and the Cochrane database was conducted to support our judgement on the APEASE criteria. We also searched for examples of existing educational interventions for OPs in the Netherlands. When literature for the context of OPs or WHS to support a judgement on the APEASE was lacking, the criteria were judged as unknown.

To reach consensus about the BCTs to be included in the intervention, a group discussion was held within the research team. First, all BCTs that met the criteria of affordability, practicability, acceptability, and effectiveness were included as they are prerequisites for implementing an intervention. Subsequently, BCTs of which the affordability, practicability, acceptability, or effectiveness was unknown were discussed by the research team, and a final decision was made on which BCTs to include in the intervention.

### 2.8. Step 8: Identify Modes of Delivery

In step 8, the most appropriate modes of delivery for the intervention were identified. Examples of modes of delivery from the taxonomy provided by the BCW framework are: at individual level or at group level, face-to-face or distance delivery, using printed, digital or broadcast media, and using computer programs or mobile phone programs [18]. Each mode of delivery from the taxonomy was judged on the APEASE criteria supported by literature. We searched for articles on group training, traditional lectures, individual training and education, and online training and education. In addition, we searched for examples of existing educational interventions for OPs in the Netherlands. When literature for the context of OPs or WHS to support a judgement on the APEASE was lacking, the criteria were judged as unknown.

Modes of delivery that met the criteria of affordability, practicability, effectiveness and acceptability were selected for the intervention. Modes of delivery of which a judgement on affordability, practicability, effectiveness and acceptability was missing were discussed on appropriateness for the intervention through group discussion in the research team. Finally examples of how these modes of delivery could be used to deliver the intervention were discussed.

## 3. Results

### 3.1. Step 1: Define the Problem in Behavioral Terms

#### 3.1.1. Stakeholders in the Implementation of WHS

The OP is an important stakeholder in the implementation of WHS, both as initiator and as implementer. Other stakeholders involved are employers, workers (i.e., works council), healthcare providers (e.g., occupational healthcare professionals), and possibly managers of OHS.

#### 3.1.2. Desired Behaviors of OPs and Other Stakeholders

The OP starts with initiating WHS at companies, explaining the added value and stressing the importance of implementing WHS for the company. To facilitate the initiation of WHS within a company, the OP needs the support of employers and workers (i.e., works council). Finally, the employer has to agree to the implementation of WHS within their company.

The OP determines the goals and content of WHS. The employer and works council and OP have to agree on the determined content of the WHS, and on the budget available. To facilitate the implementation of possible preventive interventions, agreements should be made in advance between occupational healthcare providers and the OP.

The OP conducts medical examinations and, subsequently, consultations with workers to provide feedback on the results of the medical examinations and advises on implementation of preventive interventions. If possible, the OP provides preventive advice on group level. Workers can participate by taking part in the medical examinations and in the consultation with the OP. The employer has to facilitate the implementation of preventive interventions in the workplace. Implementation of preventive interventions can be multidisciplinary and facilitated by other occupational healthcare professionals. For example, occupational physiotherapists or ergonomist are to facilitate preventive interventions related to physical health complaints, and occupational social workers or psychologist are to facilitate preventive interventions related to psychological health complaints. 

Finally, the OP evaluates the implementation of WHS with the employer and workers (i.e., works council) and makes agreements on possible follow-up.

### 3.2. Step 2: Select the Target Behavior

An earlier survey among OPs revealed that 75% of OPs have performed any WHS in the past five years. In addition, 52% of the OPs have implemented WHS no more than three times in the past five years. This finding suggests that WHS is not implemented regularly. Initiation of WHS is the first step, and it is crucial to the implementation of WHS. The initiation of WHS might be a barrier in the implementation of WHS and was selected to be targeted in the intervention.

Nearly half of the OPs who performed WHS did not discuss the results of the screening with workers, nor advised interventions. Conducting consultations with workers to provide feedback on the results of the WHS, and suggestions on preventive interventions, is crucial in protecting workers from work-related diseases and improving workers’ health and work functioning. Discussion of the results with workers and providing advice on preventive interventions might be barriers; they were, therefore, selected as behavior to target in the intervention.

### 3.3. Step 3: Specify the Target Behavior

The first target behavior is that WHS is not initiated and, therefore, not implemented. WHS should be initiated by OPs; they should advise employers about the importance and added value of the implementation of WHS at their company. The initiation of WHS is done by OPs; employers need to agree to the implementation of WHS within the company. The initiation and subsequent implementation of WHS should be done periodically, ideally before workers develop a work-related health complaint or disease. The initiation can be done in an annual meeting with the employer, held at the company, or occupational health service.

The second target behavior is that preventive consultations with workers, as part of the WHS, are not conducted. In general, the consultations are carried out by the OP, but this can also be done by occupational health nurses. OPs should discuss the results of the WHS with workers and provide them with advice on preventive interventions. Workers have a role in the consultation as they have to participate in the consultation. If necessary, the employer or healthcare provider facilitates the implementation of preventive interventions. The discussion of the results and provision of advice on preventive interventions is carried out after the surveillance has been done. Consultation can take place at the company, at the OP’s practice, or at the occupational health service.

### 3.4. Step 4: Identify What Needs to Change

The following COM-B components were selected for change through an intervention for OPs: (1) the OP has sufficient knowledge to implement WHS (e.g., to conduct preventive consultations with workers) (psychological capabilities), (2) the OP has sufficient knowledge to determine the content of WHS (psychological capabilities), (3) the OP has sufficient knowledge to initiate WHS at the employer’s (psychological capabilities), (4) the employer has a positive attitude towards WHS (social opportunities), and (5) the OP has the skills to implement WHS (e.g., to conduct preventive consultations with workers) (physical capabilities). 

During the group discussion in the last round of the nominal group technique, it was agreed that having sufficient knowledge about and being able to determine the content of WHS is necessary when explaining the importance of WHS to employers. Furthermore, it was discussed that knowledge and skills, for example, to implement WHS, are complementary and should both be included in the intervention. The selected COM-B components are shown in Table 1. A detailed report of the entire meeting can be found in Appendix A.

### 3.5. Step 5: Identify Intervention Functions

The intervention functions linked to the selected COM-B components were education, training, enablement, environmental restructuring, and modeling. Based on the APEASE criteria and group discussion among the research team, education, training, and enablement were selected as the most appropriate intervention functions to include in the intervention. The selected intervention functions, linked to the COM-B components, are shown in Table 1.

Training programs for OPs are affordable, practicable and acceptable as numerous required in-service programs for OPs already exist [22]. Effectiveness of training programs for increasing skills or self-efficacy for (occupational) physicians has been reported in literature [23]. Offering training programs to OPs to increase their skills in implementing WHS is unlikely to lead to unwanted side effects of inequity.

Educational programs for OPs are affordable, practicable, and acceptable as numerous educational programs to increase knowledge of (occupational) physicians exist already [24,25]. Moreover, education as a way to improve knowledge of OPs has proven to be effective [26]. It is also unlikely that educational programs for OPs will lead to unwanted side effects or inequity.

Enablement was selected as an appropriate intervention function for increasing opportunities for OPs to initiate WHS at companies. For example, enabling OPs by providing a helpdesk to discuss barriers they experience during initiation of WHS is affordable, practicable, and acceptable as a similar knowledge center for medical surveillance in workers already exists [27]. No research has been done into the effectiveness of this form of enablement. It is unlikely that enablement will lead to unwanted side effects or inequity.

An overview of the APEASE criteria for all listed intervention functions is shown in Appendix A.

### 3.6. Step 6: Identify Policy Categories

The policy categories that were linked to the intervention functions training, education, and enablement were: guidelines, fiscal measures, regulation, legislation, environmental and social planning, service provision, and communication/marketing.

### 3.7. Step 7: Identify Behavior Change Techniques (BCTs)

An overview of BCTs most frequently used to deliver the intervention functions training, education, and enablement, as well as judgement based on the APEASE criteria, can be found in Appendix A. The selected BCTs for the intervention are described below.

#### 3.7.1. Information about Social and Environmental or Health Consequences

The effect of providing information about health consequences of work-related risk factors, or social and environmental consequences of the implementation of WHS in different occupations to increase the knowledge of OPs is unknown. However, multiple guidance documents for OPs providing information about consequences of for example work-related risk factors exist and are affordable, practicable, and acceptable for OPs [14,28], for example, the guidance document on hand eczema [29]. Based on positive judgement on affordability, practicability, and acceptability, the research team assessed the BCTs ‘information about social and environmental consequences’ or ‘information about health consequences’ to increase the knowledge of OPs for determining the content of WHS as appropriate to include in the intervention for OPs.

#### 3.7.2. Demonstration of Behavior

To increase skills of OPs, a good example of a conversation with the employer or a preventive consultation on the individual results of a WHS with a worker can be demonstrated in an instruction video, or during a training session for OPs. Demonstration of behavior is affordable, acceptable, and practicable as it has been used before to teach skills to OPs [24].

#### 3.7.3. Instructions on How to Perform a Behavior

Instructions on paper in the form of a seven-step protocol already exist to guide OPs in conducting consultations with workers [30]. Instructions can be included in training programs for OPs; these are affordable, practicable, and acceptable [28]. We considered the BCT instruction on how to implement WHS as appropriate to include in the intervention for OPs.

#### 3.7.4. Behavioral Practice

Behavioral practice can be implemented in training programs, for example, in the form of role-play exercises using case vignettes, to practice skills for determining the content of WHS. The BCT ‘behavioral practice’ was considered acceptable, affordable, and practicable as earlier training programs also used a form of behavioral practice to improve the skills of OPs [24].

#### 3.7.5. Feedback on Behavior

Exercises, for example, role-play exercises, can be done within a training program, and OPs can provide each other with feedback, or feedback can be provided by the teacher. The BCT providing feedback on behavior is considered appropriate based on the criteria of affordability, practicability, and acceptability.

#### 3.7.6. Goal Setting, Action Planning, and Reviewing Goals

Setting goals at the end of a training program might be practicable and acceptable for OPs. After goals are set, OPs can make a plan for the actions they need to take to achieve the goals set. This can be done at the end of a training program to improve skills to initiate WHS at companies. Evaluation and reviewing goals can, for example, be done in subsequent training. We assume that goal setting may be effective, and, for this reason, we include the BCT ‘goal setting’, together with ‘action planning’ and ‘reviewing goals’, in the intervention for OPs.

### 3.8. Step 8: Identify Modes of Delivery

Based on the APEASE criteria, face-to-face contact on group level and internet-based delivery were identified as appropriate modes for delivering the intervention. An overview of modes of delivery from the taxonomy, and judgement based on the APEASE criteria, can be found in Appendix A.

#### 3.8.1. Face-to-Face on Group Level

Face-to-face delivery on group level fits in well with the identified intervention functions training and education. Face-to-face lectures are often used to inform OPs about different topics in occupational health care and are affordable, practicable and acceptable. It is an effective way to increase the knowledge of OPs [31]. Face-to-face training programs are often focused on improving skills and self-efficacy. Braeckman et al. found an increase in the knowledge and confidence of OPs after a training program focused on occupational health research and surveillance [23]. Training programs focus on improving skills and often include interactive components to practice skills; for example, role-play exercises can be included in a training program.

#### 3.8.2. Internet-Based Mode of Delivery 

Online education and training can be a valuable addition to face-to-face training programs. Based on the APEASE criteria, e-learning was identified as an appropriate mode of delivery for the intervention. Several e-learning modules for OPs already exist [32]. E-learning is affordable, practicable and acceptable for OPs. The study of Hugenholtz et al. found that an e-learning module for OPs about evidence-based medicine was effective in increasing the knowledge of OPs [33]. The e-learning in the intervention can consist of informative text, instruction texts and videos (example videos are included to demonstrate behavior), and knowledge quiz questions with standardized feedback.

#### 3.8.3. The Intervention

The proposed intervention (i.e., e-learning and training program) will consist of two modules: (1) the initiation of WHS and (2) the implementation of WHS. The e-learning should take approximately 2 h per module, and can be completed individually by OPs without the guidance of a teacher. The training program, with the guidance of a teacher, is designed to consist of two days. As there are different sectors and workplaces with different types of health risks and health complaints for workers, a basic e-learning module will be developed which can be adapted for different sectors. The criteria set for the trainer are (1) a background in occupational healthcare and (2) experience in education for physicians. The proposed details of basic the e-learning and training are described in Table 2.

## 4. Discussion

### 4.1. Summary of the Results 

The eight steps of the BCW framework were applied to systematically develop an intervention for OPs aimed at overcoming the barriers to the implementation of WHS. Behaviors targeted in the intervention were ‘OPs initiate WHS at companies’ and ‘OPs conduct preventive consultations with workers’. Knowledge and skills of OPs, and opportunities for OPs to initiate WHS were selected as behavioral components that needed change. A proposal of an e-learning and face-to-face training course was developed consisting of the following elements: providing information about the consequences of work-related risk factors for workers, demonstration of behavior, (e.g., conducting preventive consultations with workers), instructions on how to perform behavior, behavioral practice during role play exercises, feedback on behavior during role play exercises, and goal setting, action planning, and reviewing goals regarding the initiation of WHS at companies.

### 4.2. Comparison with Other Work

In this study, we used the COM-B components, capabilities, opportunities, and motivation, to identify which behavioral components needed to change. The study of Ojo et al. and Horppu et al., using the BCW framework, applied the theoretical domains framework to identify behavioral components of the problem behavior that needed change [34,35]. The theoretical domains framework is an extension of the COM-B components, and fewer options of intervention components, such as intervention functions and BCTs, are linked to each theoretical domain compared to the COM-B components [18]. Using the theoretical domains framework may be a more transparent approach to translating the targeted behavioral components into an intervention. We reasoned that the theoretical domains would not give more insight into the barriers for the implementation of WHS. For example, to investigate psychological capability, behavior can be mapped into the theoretical domains of knowledge, cognitive, and interpersonal skills, memory, attention, decision processes, and behavioral regulation, while the investigation of self-perceived knowledge about different components of the WHS provides more insight into what should be the focus of the intervention in order to improve OPs’ psychological capabilities. Moreover, a previous study conducted a validation using the theoretical domains framework and found that the use of COM-B components as opposed to the theoretical domains framework would result in the same intervention components [36]. This finding supports the use of COM-B components as sufficient for developing our intervention.

In some studies, literature have been used to explore behaviors that may cause the problem [36,37], while, in this study, only the guidance document on WHS was used. Conducting a literature search provides a broad perspective of possible behaviors that may cause the problem, and a possible perspective of behaviors that would not have been considered without the literature search. As in this study the guidance document used lists the desired behaviors for the implementation of WHS, it is possible that certain behaviors that may play a role in the problem are not taken into consideration. However, the behavior in this study is more generic than, for example, sedentary behavior of office workers in other studies, which made us reason that using the guidance document is sufficient in this study to define the problem behavior that forms the basis for the development of the intervention.

### 4.3. Strengths and Limitations 

To systematically develop the intervention, the BCW framework was selected a priori, and a survey study among OPs to investigate the problem behavior was developed based on the COM-B components, capabilities, opportunities, and motivation, of the BCW framework. As a result, the intervention is aligned with the needs of OPs, which increases the likelihood of acceptability and positive effect of the training on the knowledge and skills of OPs. Using the COM-B components theory ensured a link between the targeted behavioral components and behavior of OPs regarding the implementation of WHS. In addition, the BCW framework enforces consideration of a broad range of options in developing the intervention, and allows for transparency in choices made. For example, goal setting, action planning, and reviewing of goals would probably not have been considered for inclusion in the intervention without the use of the eight steps of the BCW framework. Using the theory-based BCW framework can, therefore, be considered a strength of this study.

Although the use of the APEASE criteria provided by the BCW framework contributes to a systematic and structured approach to the identification of appropriate intervention content, subjective and pragmatic decisions also needed to be made by the research team. To substantiate the choices made, available literature was used to support our judgement on the APEASE criteria, which may be a strength of the method used. However, the use of literature led to positive judgement of intervention types which have already been used and described in literature before, leaving less well-known and perhaps more innovative options that may be promising less likely to be selected, which may also be a limitation of this approach.

As health behavior involves multiple stakeholders, involving different stakeholders in the intervention development process is important. In our study, OPs were identified as key stakeholders [14], and results of a survey study conducted among OPs were used to investigate the target behavior for the intervention. It is a limitation of our study that the needs of employers and workers who are also involved in the implementation of WHS were not investigated. This may have led to potential opportunities for interventions to improve and increase implementation of WHS being left out.

### 4.4. Future Research and Practical Implications 

Barriers in OPs’ knowledge and skills for the initiation and implementation of WHS were identified, and selected as behavior to target in the intervention. Although barriers were identified in opportunities experienced by OPs for initiating WHS, employers were not involved in the intervention. In future studies, the role of employers and of workers in the implementation of WHS need to be investigated more extensively. An intervention to reach and convince employers of the importance of WHS might be necessary in order to increase the probability of implementation of WHS. 

The next step in this research project among OP’s is to get insight into the acceptability of the developed intervention in the current cohort study. We will assess OPs’ views on the appropriateness of the length of the proposed blended learning (i.e., e-learning and training) and preferences for modules on different sectors or types of health risks. 

The proposed intervention can be used as a starting point to develop several intervention modules focused on different branches or work-related health complaints, such as a WHS for nurses focused on mental health complaints [38], a WHS for hospital physicians [39], or WHS focused on contact eczema [29]. For example, to guide OPs in implementing a WHS for nurses focused on mental health complaints, an e-learning module will be developed with information about frequently occurring work-related mental health complaints in nurses, appropriate screening instruments, such as the four dimensional symptoms questionnaire [40] and Nurses work functioning screener [41], and instructions and information necessary for interpretation of the results. Furthermore, demonstration videos and written instructions will be provided to increase knowledge of OPs in conducting preventive consultations with workers. Skills of OPs in providing feedback and advise on preventive interventions in consultations with workers will be practiced in a face-to-face training. Role play exercises can used, guided by a trainer who can provide feedback on OPs’ behavior and guide participants in setting and reviewing goals for their own cases. 

## 5. Conclusions

The aim of this study was to develop an intervention for OPs aimed at overcoming the barriers to implementation of WHS using the eight-step BCW framework. Within the BCW framework, survey results among OPs, literature search about intervention programs for OPs, and structured group discussion were used. Lack of OPs’ knowledge and skills for implementing WHS, as well as opportunities for initiating WHS at companies, appeared to be important barriers. In order to facilitate OPs in the implementation of WHS, we identified the intervention functions education and training to increase knowledge and skills of OPs, and the intervention function enablement to improve opportunities, such as the attitude of employers towards WHS. Moreover, the BCTs information about consequences, demonstration of behavior, instructions on how to perform the behavior, behavioral practice, and feedback on behavior were identified to improve knowledge and skills, while the BCTs goal setting, action planning, and reviewing goals were selected to improve opportunities to initiate WHS through behavior of OPs. 

This study resulted in the proposal of an intervention to support OPs in the implementation of WHS by face-to-face group training, and e-learning. As different work-related risk factors and health complaints are present in different types of workers and branches, specific modules should be developed, using this proposal as a starting point. However, as the proposed intervention developed in this study is focused on OPs, barriers for the implementation of WHS in opportunities to initiate WHS at other stakeholders are not targeted directly. 

## Figures and Tables

**Figure 1 ijerph-18-01939-f001:**
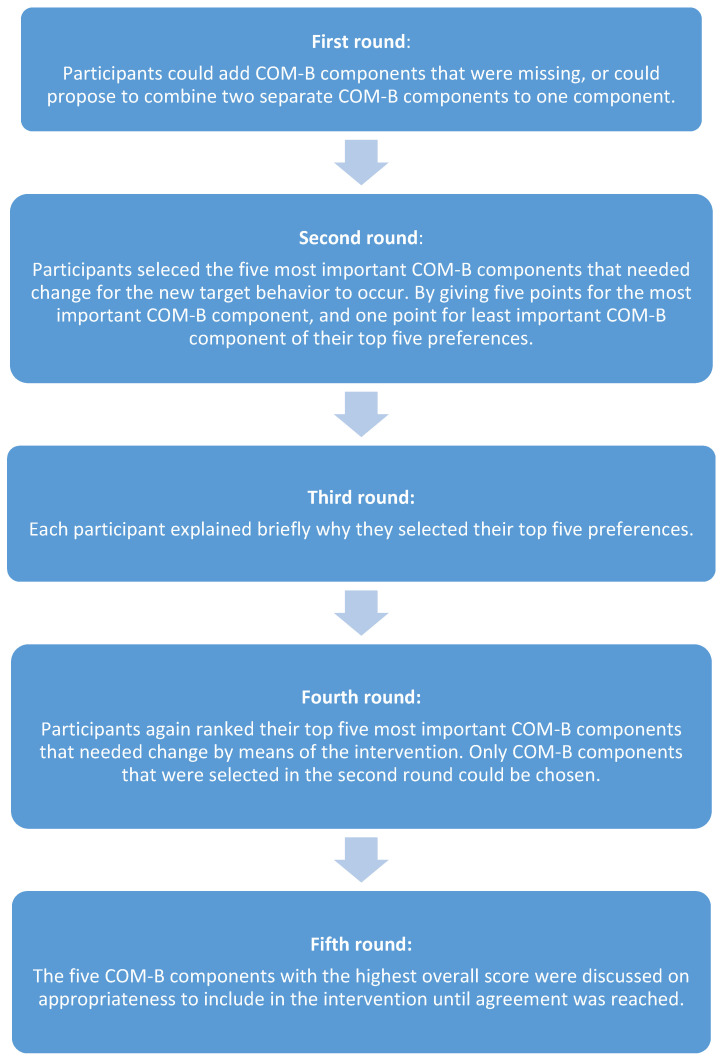
Process nominal group technique.

**Table 1 ijerph-18-01939-t001:** Combined link between behavior capability, opportunity, and motivation (COM-B), intervention functions, behavior change techniques, and mode of delivery.

Target Behavior	Desired Behavior	COM-B Components for the Desired Behavior	Intervention Function	Behavior Change Techniques	Mode of Delivery
Preventive consultations with workers are not conducted	The occupational physician is implementing workers’ health surveillance, including consultations with workers to provide feedback and advise on preventive interventions based on the results of workers’ health surveillance	The occupational physician has sufficient knowledge to implement workers’ health surveillance (e.g., to conduct preventive consultations) (psychological capabilities)	Education	Inform about social and environmental or health consequences Instruction on how to perform a behavior	Internet-based delivery
		The occupational physician has the skills to implement workers’ health surveillance (e.g., to conduct preventive consultations) (physical capabilities)	Training	Feedback on the behavior, demonstration of the behavior, instructions on how to perform a behavior, behavioral practice	Face-to face group delivery
Workers’ health surveillance is not initiated at companies	The occupational physician is initiating workers’ health surveillance at companies	The occupational physician has sufficient knowledge to determine the content of workers’ health surveillance (psychological capabilities)	Education	Inform about social and environmental or health consequences, instruction on how to perform a behavior	Internet-based delivery
		The occupational physician has sufficient knowledge to initiate workers’ health surveillance at companies (psychological capabilities)			
		The employer has a positive attitude towards workers’ health surveillance (social opportunities)	Enablement	Goal setting, action planning, and reviewing goals	Internet-based delivery Face-to face group delivery
		The occupational physician has the skills to initiate workers’ health surveillance at companies (physical capabilities)	Training	Feedback on the behavior, demonstration of the behavior, instruction on how to perform a behavior	Face-to face group delivery
		The occupational physician has the skills to determine the content of workers’ health surveillance for companies (physical capabilities)			Face-to face group delivery

**Table 2 ijerph-18-01939-t002:** Overview of the proposed intervention.

Overview of the Proposed Intervention		
Module	E-Learning	Training
Initiation of workers’ health surveillance	Informative text about work-related health risks Informative text with instructions on how to initiate workers’ health surveillance Informative text with instructions on how to determine the content of workers’ health surveillance Demonstration videos with examples of conversations with the employer to initiate workers’ health surveillance Quiz questions about work-related risk factors for workers with standardized feedback on the answers Quiz questions on which (bio)medical tests to include in workers’ health surveillance based on the workplace with standardized feedback on the answers Quiz questions about different types of resistance of employers towards the implementation of workers’ health surveillance with standardized feedback on the answers	Lecture about work-related health risks and instructions to initiate workers’ health surveillance and determine the content of workers’ health surveillance Demonstration of a conversation with the employer to initiate workers’ health surveillance Role play exercises to practice conversations with employer to initiate workers’ health surveillance, including feedback from other participants Assignment to set goals and make an action plan to reach the goals concerning the initiation of workers’ health surveillance
Implementation of workers’ health surveillance (e.g., conducting preventive consultations with workers)	Informative text about examples of preventive interventions to advise to workers based on the results of the screening Informative text with instructions on how to conduct preventive consultations with workers Demonstration videos with examples of preventive consultations with workers Quiz questions about preventive interventions to advise to workers based on the results of workers’ health surveillance with standardized feedback on the answers Quiz questions about interpretation of workers’ health surveillance results with standardized feedback on the answers	Lecture with summary of preventive interventions and instructions on how to conduct preventive consultations Demonstration of a conversation with a worker to discuss the results of workers’ health surveillance and possible implementation of preventive interventions Role play exercises to practice preventive consultations with workers and provide feedback to other participants

## Data Availability

Data is contained within the article and Appendix A. The data presented in this study are available in Appendix A.

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
