# Peer review of "Supporting Occupational Physicians in the Implementation of Workers’ Health Surveillance: Development of an Intervention Using the Behavior Change Wheel Framework"

_ijerph, 2021, doi:10.3390/ijerph18041939_

Round 1

Reviewer 1 Report

This paper presents analytically the progress for developing an intervention to support occupational physicians in implementing workers health surveillance with the use pf a behaviour change wheel framework (BCW). The topic is interesting and the manuscript is well-written. I have not much to add, but to suggest to add the table of  Appendix E (overview of e-learning and training) in the main text.

Author Response

Point 1: This paper presents analytically the progress for developing an intervention to support occupational physicians in implementing workers health surveillance with the use pf a behaviour change wheel framework (BCW). The topic is interesting and the manuscript is well-written. I have not much to add, but to suggest to add the table of  Appendix E (overview of e-learning and training) in the main text.

Response point 1: We greatly appreciate the positive comments. We agree that the table in appendix E can be added to the main text, and have added the table on page 12 and 13.

Reviewer 2 Report

The development of an occupational medicine preventive intervention methodology is presented in this study. The approach is structured using the Behaviour Change Wheel method. Although I am not familiar with the BCW as such, the methodology adopted for this work seems appropriate.

The document is very structured, easy to read and interesting. However, the details of the methodology used will be of interest primarily to academic readers in the field. In order to be of more interest to practitioners, the authors should perhaps put more emphasis on the discussion of practical tools or examples of application.

L37. Maybe "intervention" rather than "preventive measures" since health disorders are already existant

L176-177. What is the role of the other occupational health professional (e.g Occ. hyginists, ergonomist) in primary prevention in Netherlands ? They are not mentionned here.

L199-L200. You mention that OPs have performed WHS at least once in the past five years. The argument would been stronger if you mentioned a maximum frequency at which these interventions take place.

P11.L26-53. Is this module already available ? Can you give some example of similar intervention that could help the practitioners. 

Author Response

The development of an occupational medicine preventive intervention methodology is presented in this study. The approach is structured using the Behaviour Change Wheel method. Although I am not familiar with the BCW as such, the methodology adopted for this work seems appropriate.

The document is very structured, easy to read and interesting. However, the details of the methodology used will be of interest primarily to academic readers in the field. In order to be of more interest to practitioners, the authors should perhaps put more emphasis on the discussion of practical tools or examples of application.

Response: We greatly appreciate the positive comments, and have carefully revised the manuscript according to the comments and suggestions, and provided our response point-by-point, including page and line number.

Point 1: L37. Maybe "intervention" rather than "preventive measures" since health disorders are already existent

Response point 1: We agree with your suggestion and changed ‘preventive measures’ to ‘preventive interventions’ on P6,line 207; p6, line 211; p6,line 214; p7, line 250, line 251; p12, table 2.

Point 2: L176-177. What is the role of the other occupational health professional (e.g Occ. hygienists, ergonomist) in primary prevention in Netherlands ? They are not mentioned here.

Response point 2: We agree that this can be clarified, and have explained the possible role of other occupational healthcare professionals in the Netherlands in the implementation of WHS. They are mostly involved in the implementation of preventive interventions. We added this text in the results section, p6,line 216-220.

Point 3: L199-L200. You mention that OPs have performed WHS at least once in the past five years. The argument would been stronger if you mentioned a maximum frequency at which these interventions take place.

Point 3: We agree that it would be a stronger argument for development of an intervention to describe the maximum frequency of implemented WHS by OPs. We did not asked about the maximum frequency of implemented WHS. We therefore added information about how the majority of OPs in the study have implemented a maximum of three times WHS in the past five years in the results section, p6, line 225-227.

Point 4: P11.L26-53. Is this module already available ? Can you give some example of similar intervention that could help the practitioners. 

Response point 4: We have followed your suggestions to provide some examples of similar modules. We added examples of WHS modules which can be used for the content of the intervention, and described one example of an intervention content to guide OPs regarding a WHS mental health module focussed on nurses, in the discussion section,p15,line 128-144.

Reviewer 3 Report

The manuscript presented is very interesting and of enormous relevance. It is worth paying attention to how workers' health can be further supported from an occupational health perspective.

Congratulations to the authors for the comprehensive and interesting work they have presented.

Here are a number of recommendations for improving the submitted manuscript:
- It is complex to follow the thread of the abstract. Therefore, it would be interesting to order it with the classic structure: introduction, objective,....
- In the methodology, it would be interesting to include an initial paragraph explaining the process of steps to be followed.
- Finally, it would be interesting to expand the section on conclusions so that they are more in line with the objective of the work presented.

Author Response

Congratulations to the authors for the comprehensive and interesting work they have presented.

Here are a number of recommendations for improving the submitted manuscript:
point 1: It is complex to follow the thread of the abstract. Therefore, it would be interesting to order it with the classic structure: introduction, objective,....

Response point 1: We greatly appreciate the positive comments, and have followed your suggestion to structure the abstract. We have revised the abstract, and added the headings introduction, objectives, methods, results, conclusion to structure the abstract on p1, line 10-29

Point 2: In the methodology, it would be interesting to include an initial paragraph explaining the process of steps to be followed.

Response point 2: To clarify the process of development of the intervention using the BCW framework, we added a paragraph explaining the eight steps and process of the intervention development, in method section, p2, line 88-100. 

Point 3: Finally, it would be interesting to expand the section on conclusions so that they are more in line with the objective of the work presented.

Response point 3: We agree and have expanded the conclusion section by adding a more extensive answer on whether the proposed intervention may be appropriate to support OPs in the implementation of WHS by overcoming the identified barriers, in the conclusion section, p15, line 152-170.

Round 2

Reviewer 3 Report

The modifications made by the authors comply with the requirements formulated by this reviewer.